# Developing On-Site Trace Level Speciation of Lead, Cadmium and Zinc by Stripping Chronopotentiometry (SCP): Fast Screening and Quantification of Total Metal Concentrations

**DOI:** 10.3390/molecules26185502

**Published:** 2021-09-10

**Authors:** Laetitia Hackel, Elise Rotureau, Aoife Morrin, José Paulo Pinheiro

**Affiliations:** 1Laboratoire Interdisciplinaire des Environnements Continentaux (LIEC), Université de Lorraine/CNRS, UMR7360, F54501 Vandœuvre-lès-Nancy, France; laetitia.hackel@univ-lorraine.fr (L.H.); elise.rotureau@univ-lorraine.fr (E.R.); 2Insight SFI Research Centre for Data Analytics, National Centre for Sensor Research, School of Chemical Sciences, Dublin City University, Dublin 9, Ireland; aoife.morrin@dcu.ie

**Keywords:** stripping chronopotentiometry (SCP), zinc, lead, cadmium, standard addition method, screen printed electrodes, thin mercury film electrodes (TMFE)

## Abstract

Electrochemical stripping techniques are interesting candidates for carrying out onsite speciation of environmentally relevant trace metals due to the existing low-cost portable instrumentation available and the low detection limits that can be achieved. In this work, we describe the initial analytical technique method development by quantifying the total metal concentrations using Stripping Chronopotentiometry (SCP). Carbon paste screen-printed electrodes were modified with thin films of mercury and used to quantify sub-nanomolar concentrations of lead and cadmium and sub-micromolar concentrations of zinc in river water. Low detection limits of 0.06 nM for Pb(II) and 0.04 nM for Cd(II) were obtained by the standard addition method using a SCP deposition time of 180 s. The SCP results obtained for Pb(II) and Cd(II) agreed with those of inductively coupled plasma mass spectrometry (ICP-MS). The coupling of SCP with screen-printed electrodes opens up excellent potential for the development of onsite speciation of trace metals. Due to the low analysis throughput obtained for the standard addition method, we also propose a new, more rapid screening Cd(II) internal standard methodology to significantly increase the number of samples that can be analyzed per day.

## 1. Introduction

Trace metal speciation and bioavailability in natural samples [1] are key to understanding the metal toxicity and, ultimately, fate in the environment [2] and providing the basis for the regulatory water quality criteria [3], which, in turn, has a major impact on the industrial effluent’s purification cost.

Electrochemical stripping techniques are commonly used to carry out in situ or onsite quantifications of trace metals [4] due to the availability of low-cost and portable instrumentation and the very low detection limits that can be achieved. They are amongst the very few techniques that can provide trace metal dynamic and equilibrium speciation information at the environmentally relevant concentrations of metal ions [5]. These techniques comprise two steps, consisting of a first step (deposition step) to preconcentrate the target metal ion on the electrode and a second step (measuring step) where this metal ion is quantified [6].

In ecotoxicology studies only, the free metal ion (or hydrated cation) is considered toxic. Nevertheless, there are very few techniques able to quantify free metal ions, especially at the very low concentration levels present in environmental waters. Due to particular matrix effects inherent to natural samples, some precautions in the methodological choice are needed to ensure the measurement quality and performance, as explained hereafter.

Natural organic matter (NOM) is ubiquitous in natural waters and constitutes a severe problem in traditional stripping techniques like Anodic Stripping Voltammetry (ASV), since it is adsorbed onto the electrode surface [7] and strongly distorts the voltametric signal [8]. One of the few commercial systems that uses electrochemical techniques to perform environmental trace metal monitoring is the Idronaut ^®^ Voltammetric In-Situ Probe (VIP). This sophisticated system performs several stripping electrochemical techniques to quantify multiple metal ions using microelectrode arrays [9]. This is quite an expensive device both regarding the VIP and the microelectrode array and, thus, can only be used by trained researchers. Moreover, the use of such monitoring instruments is constrained by the hydrological conditions (water flow and depth).

An alternative to protecting the working electrode from organic matter fouling is to use Stripping Chronopotentiometry (SCP), a technique proposed by Town and Van Leeuwen in 2000 [10] that uses electrolysis at a constant current for the measuring step to overcome organic matter surface adsorption problems [11]. Additionally, SCP is less affected by the inter-metallic species formation common in multi-metal systems, since the slow rate of oxidation during depletive SCP measurements fosters the redissolution of such compounds within the mercury electrode [12]. For in-depth metal speciation investigations, one of the major advantages of using SCP to quantify the total metal concentration is that the same experimental setup can be used sequentially to evaluate the free metal ion using the Absence of Gradients and Nernstian Equilibrium Stripping (AGNES), a technique performed under equilibrium conditions [13], and Scanned Stripping Chronopotentiometry (SSCP) to obtain dynamic information regarding the heterogeneity of the metal complexes [14].

Previously, we have proposed a methodology for Cd(II) speciation by SCP, AGNES using thin mercury films (TMF) on carbon screen-printed electrodes coupled with ultrafiltration [15]. The obtained limit of detection (LOD) for the total metal was 1.6 nM, still considerably higher than the natural Cd(II) concentration in the river water tested. Thus, it was necessary to spike the samples prior to analysis. Therefore, there is a clear need to develop new methodologies that can effectively measure trace levels of target metal ions in the environment without spiking with high sensitivities.

Briefly, one of our main objectives is to perform on-site trace metal speciation at the concentration levels present in natural waters. To achieve this goal, first, it was necessary to quantify the total metal concentration at the appropriate levels, which we describe in this paper. In the future, the second step will be to quantify the free metal concentration by the Absence of Gradients and Nernstian Equilibrium Stripping (AGNES) in the pristine sample using the same electrochemical system (same cell, electrodes and instrument).

The third step will be to investigate the dynamic speciation of the trace metal using the unique properties of Scanned Scanning Chronopotentiometry (SSCP) again in the same electrochemical system. SSCP will be carried out in the same sample solution as the free metal determination, while the total metal concentration will be carried out independently, since the sample solution is strongly acidified.

As the free metal ion concentration is much lower than the total metal concentration, free metal quantification and dynamic speciation studies will not be possible in cases where the total metal concentration is very low, i.e., close to or below the LOD for the total metal. Thus, we are interested in developing a rapid screening technique that will help decide which samples require speciation studies and those in which the total metal concentration is too low to justify a further analysis. For this screening, we propose the Cd(II) internal standard methodology, which we report here.

Another motivation to develop total trace metal quantification in field systems arose from our participation in a project to develop novel geochemical exploration technologies (NExT) that explored the anomalies in total trace metal concentrations in different surface sources (soil horizons, plant tissues, snow, etc.) in order to locate the subsurface mineralization using statistical tools for the reliable interpretation of elemental spatial anomaly patterns [16]. Trace metal ion quantification in environmental water matrices is not usually carried out in the field but in certified analytical laboratories, most notably using the inductively coupled plasma optical emission spectroscopy (ICP-OES) or ICP-mass spectrometry) (ICP-MS) techniques. This quantification provides the metal concentration after the sampling, transport to the laboratory, storage and pretreatment. The later consists commonly of filtration through either a 0.22 or 0.45 µm filter and strong acidification of the sample (usually to pH ≤ 1). This pretreatment is also sometimes performed in situ, several minutes after sampling.

The main advantage of lab quantification techniques is that multiple metal ions can be quantified simultaneously at a low (OES) to very low (MS) detection limit, while the use of automatic samplers allows large numbers of samples to be analyzed per day. The disadvantages are that these analyses tend to be executed several days to several weeks after the sampling, are globally expensive, especially if we are only interested in one or a few metal ions, and finally, no information on the metal ion speciation can be obtained, since only the total metal concentration is quantified using these lab quantification techniques. Therefore, a clear interest in measuring trace metal ions in the field by electrochemical techniques exists, as the latter can enable the in situ monitoring of trace metal ions with lower sample handling, faster sample-to-answer times and lower cost measurements.

Therefore, the objectives of this work were to develop a low-cost, in situ SCP analytical method to (i) determine the total metal concentrations with a sub-nanomolar LOD for Cd(II) and Pb(II) and sub-micromolar LOD for Zn(II) in real environmental matrices and (ii) increase the analysis throughput per day beyond what is typically achieved (three to five analyses per day) using these types of techniques.

## 2. Theory

Metal detection is achieved using stripping chronopotentiometry (SCP). As depicted in Figure 1, SCP comprises two steps.

The first one applies a deposition potential (*E*_d_) in the limiting current region for a deposition time $t_{d}$ under stirring conditions. The second step involves the reoxidation of the amalgamated metal imposed by the application of a stripping current, *I*_s_, which lasts until the potential reaches a value well past the reoxidation transition plateau. The resulting electrolysis time, *τ**, allows the quantification of the total accumulated metal. This time is measured by the peak area obtained by plotting the derivative d*t*/d*E* vs. *E*.

The relationship between *τ** and the metal concentration in the solution is explained hereafter. During the deposition step, the number of moles deposited in the mercury electrode equals:(1)$$N_{\mathrm{deposited}}=\frac{I_{d}^{*}t_{d}}{nF}$$
where $I_{d}^{*}$ is the limiting current, *n* is the number of exchanged electrons and *F* the Faraday constant.

During the stripping step, the number of moles re-oxidized is given by:(2)$$N_{\mathrm{oxidised}}=\frac{I_{s}\tau^{*}}{nF}$$
where $I_{s}$ is the stripping current, and $\tau^{*}$ the limiting value for the transition time. Therefore, the charge balance for complete depletion is:(3)$$\tau^{*}=I_{d}^{*}t_{d}/I_{s}$$

In a non-complexing medium, such as acidified solutions, the following relationship between the limiting current and total metal concentration applies:(4)$$I_{d}^{*}=\frac{nFAD_{M}c_{M,T}^{*}}{\delta }$$
where *D*_M_ is the metal diffusion coefficient, $c_{M,T}^{*}$ is the total metal concentration in the bulk solution and δ is the diffusion layer thickness given by:(5)$$\delta =kD_{M}^{p}$$
where k depends on the geometry of the electrode and the hydrodynamic conditions of the cell, and *p*-varies between 1/2 and 2/3 [18].

Standard linear calibration plots are employed to investigate the LOD and limits of quantification (LOQ) of SCP, while the standard addition method is used to quantify the metal ions present in the river samples due to its ability to eliminate the matrix effects.

The limit of detection (${\mathrm{LOD}}_{S_{y}}$) and limit of quantification (${\mathrm{LOQ}}_{S_{y}}$) for the calibration method are related to the standard deviation of residuals (*s*_y_) in electrochemical measurements, given by [19]:(6)$${\mathrm{LOD}}_{S_{y}}=\frac{t\left(v,\alpha \right)s_{y}}{m}\;;{\mathrm{LOQ}}_{S_{y}}=\frac{3\;t\left(v,\alpha \right)s_{y}}{m}$$
where *m* is the slope of the calibration plot, and *t*(*v*,α) the Student’s *t* parameter for a probability *v* and degree of freedom α.

The calibrations were performed measuring in triplicate the blank and four aliquots for a total of 15 measures. The Student’s parameter for 99% is 3.012; thus, we decided to use the simplified limits proposed by Ziegler [20], even if they are slightly smaller than the real values:(7)$${\mathrm{LOD}}_{S_{y}}=\frac{3s_{y}}{m}\;;{\mathrm{LOQ}}_{S_{y}}=\frac{10s_{y}}{m}$$

The reverse-axis standard addition analysis is used [21] to compute the concentration from the y-intercept and, thus, obtain the error estimation and the LOD from the respective standard error (*s*_b_).
(8)$${\mathrm{LOD}}_{S_{b}}=3s_{b}\;;{\mathrm{LOQ}}_{S_{b}}=10s_{b}\;$$

## 3. Results

In this section, we examined the issue of lowering the detection limit for the SCP technique; then, we tested the internal standard methodology in order to validate this experimental strategy needed to increase the number of SCP measurements per day. Finally, the last part is dedicated to the comparison between the river sample results issue from our SCP methods and those obtained from ICP-MS.

### 3.1. Detection Limits for Cd(II), Pb(II) and Zn(II) for Calibration and Standard Addition Methods

To study the LOD for the three metals, we first performed a set of simultaneous calibrations at different deposition times and calibration ranges in 10 mM NaNO_3_ (pH 3.5) (Table 1).

By increasing the deposition time from 45 s to 240 s, we expected to significantly improve the LOD as the analytical signal *τ** is directly proportional to *t*_d_ (Equation (3)). From Table 1, it is clear that the LOD depends more on the opted calibration range than on the time deposition potential. The average LODs for a deposition time of 45 s using a calibration range in the interval 5–75 nM are 1.9 ± 0.5 nM for Zn(II), 1.6 ± 0.5 nM for Cd(II) and 1.4 ± 0.5 nM for Pb(II). This LOD is adequate for natural waters, whereby the Zn(II) concentrations are generally greater than 10 nM. However, it is not low enough to quantify Pb(II) nor Cd(II), for which the normal levels in unpolluted water are generally below 1 nM and 0.1 nM, respectively. By decreasing the calibration range to between 0.5 and 4.0 nM and increasing the deposition time to 180 or 240 s, we observed LODs of 0.044 ± 0.009 nM for Cd(II) and 0.072 ± 0.007 nM for Pb(II), which are sufficiently low to meet our demands.

To confirm these results obtained, we carried out a series of standard addition calibrations for Pb(II) and Cd(II) in acidified natural river samples (both at pH 3.5 and pH < 2).

Table 2 gives the results obtained by the standard addition method and shows that, in the river water samples, the LOD*s*_b_ obtained is slightly lower than those obtained for the laboratory calibrations, since, normally, *s*_b_ is smaller than *s*_y_. The results vary between 0.02 and 0.07 nM for both Cd(II) and Pb(II), which is satisfactory for Pb(II) but not for Cd(II), as many natural waters have concentrations similar to the reported LOD. Another interesting aspect is that there are no significant differences (within the same margins) between the LODs obtained at pH 3.5 and at pH < 2.

### 3.2. Fast Screening INTERNAL Standard Method Using Cd(II)

The internal standard method consists of adding a known amount of a certain analyte that is absent or below the LOD in the sample and use it to determine the concentrations of the other analytes present. In our case, we proposed to use Cd(II) as the internal standard, since, in natural waters, its concentrations are usually equal or below the LOD obtained here, and in any case, they are smaller than those of Pb(II) and much smaller than those of Zn(II).

To use this method, it is necessary to know exactly the proportionality between the SCP calibration slopes of the Cd(II) case and those of Pb(II) and Zn(II). According to Equations (4) and (5), the SCP signal should be equivalent for three metals, with the only systematic difference being due to their different diffusion coefficients, to a power, *p*, that varies between 1/2 and 2/3, depending on the stirring and electrode geometry [18]. Thus, the theoretical Pb/Cd and Zn/Cd ratios should be equal to (*D*_Pb_/*D*_Cd_)^p^ and (*D*_Zn_/*D*_Cd_)^p^, respectively.

Examining the evaluation of the diffusion coefficient of metal ions in aqueous solution by Kariuki and Dewald [22] in 1:1 electrolytes at I = 0.1 M, we observed that nine values were reported for Cd(II), with an average value of 7.12 × 10^−10^ m^2^ s^−1^ in the interval 6.93 to 7.41 × 10^−10^ m^2^ s^−1^, five values were given for Pb(II), with an average value of 8.90 × 10^−10^ m^2^ s^−1^ in the interval 8.28 to 9.85 × 10^−10^ m^2^ s^−1^ and, finally, three values were provided for Zn(II), with an average value of 6.67 × 10^−10^ m^2^ s^−1^ in the interval 6.38 to 6.90 × 10^−10^ m^2^ s^−1^. Taking all possibilities into account, the Pb/Cd and Zn/Cd calibration slope ratios should be contained in the intervals 1.08–1.24 and 0.93–0.98, respectively.

To validate the theoretical ratios, we carried out a series of simultaneous SCP calibrations for the three metals, using a reduction potential (*E*_d_) set at −1.35 V. The experiments were conducted in 20 mL aliquots of solution with an electrolyte concentration of 10 mM NaNO_3_ (pH 3.5). This pH is the lowest that can be used in Zn(II) quantification due to the interference of hydrogen evolution at the working electrode at this deposition potential.

Table 3 depicts the results of these calibrations, and the average and standard deviation for the Pb/Cd ratio is 1.3 ± 0.1, while, for the Zn/Cd ratio, is 0.68 ± 0.06.

Comparing the theoretical and experimental ratios, we observed that the measured Pb/Cd was within the error of the theoretical estimation (1.08–1.24), while the measured Zn/Cd was clearly lower than expected (0.93–0.98). This may be due to the fact that the Zn(II) reduction at the mercury electrode was not fully reversible [23], which tended to decrease the analytical signal. Nevertheless, there was good repeatability for the ratios.

While previous results were carried out in well-known solution compositions, the validation of the hypothesis should be carried out in natural water samples. Hence, we performed a series of standard additions in river water samples acidified to pH 3.5, measuring the three metals using a deposition time of 90 s and deposition potential of −1.35 V at the natural ionic strength of the sample (Table 4). The average and standard deviation for the Pb/Cd standard addition slope ratio was 1.1 ± 0.1, which was still in agreement with the range of possible theoretical values (1.08 to 1.24), although smaller than observed for the calibrations in simple electrolyte solutions. This may be due to a difference of ionic strength or, most likely, to some matrix effect as a result of the presence of natural organic matter (NOM).

The experimental Zn/Cd slope ratio obtained in the river water samples was 0.28 ± 0.06, which was significantly smaller than that observed in the previous standard calibration experiments (0.68 ± 0.06). Eliminating the two extreme values (0.09 and 0.48), we got the same average but a much smaller standard deviation (0.28 ± 0.03) that, in terms of the relative error, was 11% (0.03/0.28 × 100), similar to the 9% found for the calibrations (0.06/0.68 × 100). The decrease in the ratio originates from the smaller Zn(II) calibration slopes observed for the case of river waters, which, again, may be due to changes in the reversibility or some matrix effect.

The important point to note is that the ratios are reasonably stable, with standard deviations (or relative errors) similar to those obtained in the calibration experiments, which suggests that the ratio obtained in river water samples can be used for internal standard experiments. Considering this encouraging result, we compared the data obtained by the standard addition and internal standard methods in river sample measurements to further investigate the feasibility of the internal standard method.

### 3.3. Determination of Pb(II) and Zn(II) in Real Samples by Standard Addition and Cd(II) Internal Standard and Their Comparison with the ICP-MS Results

When developing new analytical methods, it is important to validate the results with a reference method. In this section, we compare the standard addition and internal standard method with ICP-MS measurements performed on the same samples. Before proceeding with the results, it is interesting to provide an actual example of a standard addition measurement and the respective reversed-axis standard addition method to illustrate our data analysis and error estimation. Figure 2 shows the initial sample SCP curve at five standard additions for Cd(II) and three standard additions for Zn(II) and Pb(II) (one result is not shown for the sake of clarity). We recall that the analytical signal is the area under the peak and corresponds to the electrolysis time (*τ**) of Equation (3).

The Zn(II) signal corresponds to the peak at −0.95 V, Cd(II) to the peak at −0.55 V and Pb(II) to the peak at −0.37 V, and it is quite clear that there is a significant initial amount of Zn(II), while none is observable for either Cd(II) or Pb(II). Another aspect is that both Cd(II) and Pb(II) peak areas increase more than the Zn(II) with similar additions.

The internal standard procedure is carried out using the addition of Cd(II) corresponding to the red line from Figure 2. Firstly, the three peaks areas are measured. To obtain the Zn(II) concentration, we first divide the Zn(II) peak area by the Cd(II) peak area, multiply this value by the Cd(II) concentration and divide the result by 0.21 (the Zn/Cd slope ratio given in Table 3). This gives a result of 31 ± 5 nM (Table 5). Instead of using the Zn/Cd day value of 0.21, we can alternatively use the average value 0.28 (± 0.06) from our results of Table 4 to obtain a concentration of 24 ± 5 nM (Table 5).

A similar procedure was applied for Pb(II), and a value of 0.4 ± 0.2 (Table 6) was found when using the day Pb/Cd ratio of 1.12 (Table 3) and 0.4 ± 0.3 (Table 6) when using the Pb/Cd average ratio of 1.09. If, for the Zn(II) values, the errors are acceptable, it is clear that, for the Pb(II) values, the errors are quite large, which is not unexpected, since the LOD for this experiment is 0.2 nM (Table 6).

Figure 3 shows the reversed-axis standard addition of the SCP peak areas (in seconds) vs. the standard addition (nM) corresponding to the SCP curves presented in Figure 2. The initial sample concentrations were derived from the y-intercept, where we can observe a value of 28 nM for Zn(II) (28 ± 1 nM from Table 5), a very low value for Pb(II) (0.3 ± 0.1 nM reported in Table 6) and a value of 0 for Cd(II).

There is a good agreement between the standard addition method and the internal standard method; nevertheless, the latter does have substantially large errors. The reported errors for the standard addition method are the standard deviations at a 95% confidence interval using the *s*_b_, while the internal standard errors are calculated using error propagation applied to the computation.

Now that we have a better vision of the data treatment and SCP concentration calculations, let us observe the experimental results and their comparisons with the ICP-MS measurements. First, let us analyze the Zn(II) results from SCP, presented in Table 5. We recall here that the solution pH cannot be lowered below 3.5 due to interference from hydrogen evolution at the working electrode. For this set of data, the LOD is much smaller than the values present in the river waters.

There is a very good agreement between the internal standard concentration obtained with the day’s Zn/Cd ratio and a worse agreement with the average Zn/Cd ratio value. Again, the standard addition error is significantly smaller than the one derived from the internal standard.

The results found for several unfiltered Zn(II) samples showed that the majority amongst them provided concentration values larger than those determined from the ICP-MS results. This discrepancy seems logical, since the acidification of unfiltered samples will probably lead to a release of the complexed or adsorbed fraction of Zn(II) to mineral particles, natural organic matter colloids or living organisms that were otherwise retained in the filter and, thus, undetected by the ICP-MS.

The most unexpected result is that all filtered samples analyzed by SCP, except one (Bethdwn 20.02.03), tend to give significantly lower concentrations, i.e., one-third to one-half times lower than the ICP-MS results. This may be due to the differences in the acidification protocols, as ICP samples have a pH < 2 while the Zn(II) SCP measurements are carried out at pH 3.5.

Now, let us focus on the Pb(II) case. We analyzed the same unfiltered samples that were investigated for Zn(II), and the results are presented in Table 6. The observation for Zn(II) also applies to Pb(II).There is a good agreement between the concentrations derived from the internal standard and standard addition methods, and eleven of the fourteen SCP concentrations were larger than those reported by the ICP-MS, even if, in the Pb(II) case, the concentrations measured were much closer to the LOD.

We also performed the analysis of the filtered samples for Pb(II) successively with three different methods (Table 7): at the deposition time 90 s and acidified to pH 3.5, at deposition time 180 s and acidified to pH 3.5 and, finally, at deposition time 180 s and acidified to pH < 2.

Table 7 shows that filtered samples measured at 90 s are generally below the quantification limit. Nevertheless, four of the six measures agree with the ICP-MS measures, and the standard addition results are equivalent with those from the internal standard procedure within the experimental error.

When these results were repeated at a deposition time of 180 s with the experimental conditions, providing adequate LOD (below 0.1 nM), the results match well with the ICP-MS ones. There is no systematic difference between the solutions acidified at pH 3.5 and below pH 2. The ICP-MS results provided by the two different laboratories are not always in agreement. The two methods of internal standard and standard addition are consistently in good agreement as previously observed.

Finally, we present a series of eight measurements performed solely for Pb(II) using the standard addition method, where we observed that five of them are consistent with the ICP-MS (within experimental error), while one is relatively close (0.41 ± 0.05 vs. 0.25 ± 0.08 nM) and two are clearly different (0.54 vs. 0.16 and 0.32 vs. 0.14 nM).

Taking the twenty standard addition experiments carried out with a deposition time of 180 s for Pb(II) presented in Table 7, one obtains an average LOD of 0.06 nM for our SCP methodology. This meets the criteria for Pb detection in natural waters and electroanalytical measurements that provide comparable concentration values with the ICP-MS technique.

Although Cd(II) was used as the internal standard in the development of this method; we also had the opportunity to quantify this metallic element during a short period using the standard addition method, since there was a detectable Cd(II) concentrations in the river. This is due to the excellent average detection limit of 0.04 nM obtained for the Cd(II) standard additions (Table 8). Both the sample concentration and the LOD are presented in Table 8. For five of the seven measurements, there is an excellent agreement with the ICP-MS results, and for the other two; one is overestimated (0.14 vs. 0.06 nM) and one underestimated (0.09 vs. 0.30 nM). It is interesting to note that, unlike the Pb(II) case, Cd(II) quantification performed for the same sample at pH 3.5 and <2 shows that lowering the pH increased the amount of Cd(II) detected.

## 4. Discussion

One of the major objectives of this work was to demonstrate that, with an adequate parameter choice—namely, deposition time and selection of concentration range for the calibration or standard addition methods—it was possible to lower the LOD for Pb(II) and Cd(II) to sufficiently low values so that SCP can be used to carry out speciation studies of these metal ions in natural waters. The results presented in Table 1 and 2 show that LODs < 0.07 nM were obtained for both Cd(II) and Pb(II) both in calibration standard solutions and in natural river water samples. These results meet our objective and are significantly better than those reported previously by applying similar techniques [1].

The validation of the SCP results by comparison with the ICP-MS is satisfactory for both Pb(II) and Cd(II), while, for Zn(II), the SCP method yielded a systematic under-estimation of the measured concentration. Most likely, this is due to the fact that the pH used for Zn(II) quantification cannot be lower than 3.5. Alternatively, Cu(II) present in the solution may interfere with the Zn(II) due to the formation of intermetallic complexes in the amalgam [24]. This aspect needs further investigation.

Another objective of the work was to use develop a fast screening method using an internal standard to significantly increase the number of measurements performed daily. In the results section, we have shown that using a slope ratio (Zn/Cd or Pb/Cd) obtained daily using standard addition, provides an internal standard concentration result identical (within experimental error) to that obtained by the conventional standard addition method, albeit experimental error is two to three times larger and, hence, has a lower precision. However, the proportionality between the Cd(II), Pb(II) and Zn(II) SCP signals was not the same in the calibration solutions and in the river samples. Nevertheless, in the different rivers samples, the proportionality is maintained; thus, we propose to adopt this internal standard method when measuring similar natural samples, consisting of performing a standard addition method on the first sample of the day (in order to calculate the Zn/Cd and Pb/Cd ratios), followed by the internal standard method for remaining samples for the day.

To improve the productivity, it is important to understand the different variables that affect the analysis time per sample when using the standard addition method in the field. As an example, we take as a reference the Rombas-Rajapalot field trip (Rajapalot, Finland, June 2019) (JGE in prep). The first constraint is the working electrode preparation (takes approx. 45 min) and the number of samples that can be measured per working electrode. After extensive testing, we concluded that, after four measurements, there was a significant degradation of electrode performance since, for our current field electrochemical set-up, we physically removed the electrodes from the solution at each sample change. Therefore, it is necessary to prepare two working electrodes per day, thus 1.5 h per day.

The analysis requires 15 min of purging to decrease the dissolved oxygen level in the solution (as this interferes strongly with the SCP reoxidation measurement by chemically reoxidizing the amalgamated metals [10]). Following this, the SCP measurement is performed in triplicate for the sample, and subsequently, the four x standard addition samples are also made in triplicate (5 × 3). Each of these comprises a deposition time of 180 s, to give a total analysis time of 45 min (15 × 180 s). Cleaning the electrochemical cell and changing samples takes an additional 10 min to give a total analysis time per sample of approx. 70 min, leading to 7.5 h of experimental work in the field, excluding travel and setting up time for this method.

Applying the new internal standard method means that the first sample of the day still takes 70 min but that the following samples are reduced to the purge time (15 min), the internal standard measurements in triplicate, which takes 9 min (3 × 180 s) and 10 min for cleaning the cell and changing the sample to achieve a total of approx. 34 min per analysis, i.e., half the original time per analysis. At this point, increasing the number of samples would also mean that three working electrodes would be necessary per day (2.25 h preparation time) with 10 samples at 34 min (5.67 h), leading to approx. 8 h of experimental time, excluding travel and experimental setting-up time.

It is now evident that, although using the internal standard method can double the number of analyzed samples per day, the productivity remains relatively low, of the order of 10 samples/day. To address this issue, we already proposed a new system were the working electrode is combined with an oxygen filter to overcome the need for purging the solution with N_2_ [25]. However, this system is not yet ready for the field, since it requires a second potentiostat for the operation, and the miniaturization of the coupling of the metal-sensing electrode and oxygen filter is under development.

We are also addressing the number of samples measured per working electrode by designing a new flow cell and pump system that will allow replacement of the sample in contact with the working electrode without exposing the latter to the atmosphere and decrease the cleaning time of the cell/replacement of the sample to approx. 5 min. For obvious reasons, the development of this new cell will be coupled with the new oxygen filter. In our laboratory, we have tested the same type of thin mercury films on screen-printed electrodes and observed that they work well for around 200 measurements over a period of 12 h in the same solution with excellent repeatability.

In this work we demonstrated that SCP can provide a sufficiently low LOD to quantify the total Pb(II) and Zn(II) in river samples. Cd(II) can be measured in the rare occasions when the natural concentrations rise above 0.1 nM. Demonstrating good potential regarding the future development of speciation studies of these metal ions. Additionally, the rapid internal standard screening method has been demonstrated to increase the daily analysis throughput and so will be useful as a screening tool for quickly identifying the samples requiring in-depth speciation studies.

## 5. Materials and Methods

### 5.1. Instruments

An Ecochemie Autolab type III potentiostat controlled by GPES software version 4.9 (Ecochemie, The Netherlands) was used in connection with a Metrohm 663VA instrument. Ag/AgCl Dri-ref-5 electrode from WPI (Sarasota, FL, USA), and a glassy carbon electrode (Metrohm) were utilized as a reference and counter electrode, respectively. The working electrode was a thin mercury film (TMF) plated on a screen-printed electrode composed of carbon pasting (Gwent, UK, C10903D14) (Figure 4).

The temperature and pH were measured with a pH meter, WTW pH 197. The combined pH electrode Sentix 41-3 was used for the measurement of pH and contains a PT 1000 Temperature probe.

A nitrogen cylinder connected to a gas purger provided the nitrogen necessary to eliminate the oxygen from the electrochemical cell.

### 5.2. Reagents

The Zn (II), Pb (II), Cd (II) and Hg (II) solutions were obtained from certified standard solutions at 1000 mg/L (Fluka Trace Select). Three different stocks solutions of Zn, Pb and Cd were prepared from cascade dilutions in order to obtain final concentrations of 10^−4^ M, 10^−5^ M and 10^−6^ M.

The ionic strength was set using a solution of 1 M NaNO_3_ prepared from solid powder (Sigma-Aldrich, supra-pur, Saint Louis, MO, USA). The pH was adjusted with 1 M NaOH standard (Merck titripur^®^, Darmstadt, Germany) or 1 M HNO_3_ diluted from a 65% (Merck suprapur) solution.

For the working electrode pretreatment, a cleaning solution with 0.2 M H_2_SO_4_ (Sigma-Aldrich, p.a, Saint Louis, MO, USA). A second cleaning step in a solution containing 1 M NH_4_CH_3_COO from the salt (Sigma Aldrich p.a) diluted in 0.5 M HCl (Merck, p.a, Darmstadt, Germany) was also carried out. To prepare the solution for the redissolution of the mercury film, 0.1 M NH_4_SCN from the salt (Sigma Aldrich, p.a) was used. Ultrapure milli-Q water (resistivity 18.2 MΩ cm, Elga, labwater, High Wycombe, UK) was employed in all the experiments. Nitrogen (>99.999% pure) for the solution purging was purchased from Air liquid, France.

### 5.3. Fabrication of the Carbon Screen-Printed Electrodes

Screen printing was performed with a semi-automated DEK 248 (Weymouth, UK). A nylon screen with a mesh thickness of 77 T (filaments per cm) was employed. Electrodes were screen-printed onto a preshrunk polyethylene terephthalate (PET) substrate (175 mm thickness). Carbon paste ink (C10903D14, Gwent Electronic Materials Ltd. (Pontypool, UK)) was applied and cured at 150 oC for 15 min. Two consecutive prints of the carbon were performed. A nonconductive dielectric polymer layer (Electrodag 452SS, Acheson, Port Huron, MI, USA) was applied to define the electrode area (3 mm diameter) and cured in the UV curing machine (UV Process Supply, Inc., Cortland, Chicago, IL, USA) for 3 cycles. UV lamp intensities of 300 watts per inch were employed for insulating layer curing procedures.

### 5.4. Preparation of the Working Electrode

The preparation, use and post-treatment of TMF modified screen-printed working electrodes comprised four steps, namely, (a) the pretreatment [26], (b) conditioning of the carbon paste surface, (c) mercury deposition and d) mercury recovery [27] detailed below:(a)Electrode pretreatment consisted of a linear scan voltammetry sweep between −1.2 V and +1.5 V (vs. Ag/AgCl), using a scan rate of 0.1 V/s in H_2_SO_4_ (0.2 M) to remove any impurities from the electrode surface resulting from the fabrication process;(b)the conditioning consisted of applying 50 successive cyclic voltammetry cycles between −0.8 V and +0.8 V (vs. Ag/AgCl) at 0.1 V/s in NH_4_CH_3_COO (1 M)/HCl (0.5 M);(c)for the electrodeposition of the TMF, the electrode was immersed in a Hg(NO_3_)_2_ solution 0.18 mM (pH adjusted at 1.9), and a potential of −1.3 V (vs Ag/AgCl) was applied for 420 s. Following this, a square-wave voltammetry cycle was carried out (−1.3 V to −0.2 V, amplitude 25 mV, step 4 mV, frequency 25 Hz) to verify if the response corresponded to a good quality mercury film. If this voltammogram was not adequate, it was discarded, while if the voltammogram was satisfactory, the working electrode was deemed ready to be used;(d)at the end each day, the mercury must be recovered for proper disposal. This step also provides information on the amount of deposited mercury for purposes of quality control. Mercury reoxidation was carried out in a 5 mM ammonium thiocyanide (pH 3.4) using a linear scan voltammetric sweep from −0.1 V to +0.8 V at 0.005 V/s. The integral of the peak area multiplied by the scan rate provides the re-oxidized charge that was used to compute the amount of surface deposited mercury during the step (c). Scanning electron microscopy images of similar electrodes show that they are an assembly of nano-hemispheres of mercury that works as an array of microelectrodes with superposing diffusion layers [27].

### 5.5. Experimental Protocol

A disposable polystyrene beaker was used as the electrochemical cell supported by a glass container. Polystyrene is the best material to perform metal ion quantification, since there are no losses of ions at the container walls, regardless of the solution pH [28]. Solutions were initially purged with nitrogen for 15–20 min, and a constant nitrogen flow was maintained during the experiments.

The stripping chronopotentiometry (SCP) measurements were carried out in 20 mL of initial aliquots, and the final volume never exceeded 22 mL. All added volumes were duly noted, and the dilution effects were considered in all calculations.

The metal deposition (amalgamation) at the electrode was performed by the application of a reduction potential (*E*_d_) for a chosen deposition time (*t*_d_) under agitation by means of a mechanical stirrer at rotation speed of 1000 rpm. The quantification of the amalgamated metal (stripping step) was achieved using a chronopotentiometric method by applying a reoxidation current fixed (*I*_s_) of 2 µA.

The reduction potential (*E*_d_) was adjusted according to the different types of metals, −1.15 V for Zn(II), −0.65 V for Pb(II) and −0.75 V for Cd(II), while for dual detection, the more negative potential was applied. The deposition time used depends on the desired detection limit (LOD), and in this work, values varying from 45 to 240 s were used. Between each SCP measurement, a continuous potential was applied in order to prevent ion adsorption at the electrode surface. All measurements were carried out at room temperature (21–25 °C).

The LOD, LOQ and intercalibration studies were obtained from a series of calibration measurements carried out in 10 mM NaNO_3_ (pH 3.5) for concentration ranges 3–500 nM for Zn(II): 0.5–300 nM for Pb(II) and 0.5–25 nM for Cd(II). All measurements were performed in triplicate.

River water samples were collected in the Orne river in Lorraine (Tributary of the Moselle in the North Eastern France) in three different sites: the Beth dam site (Bethdwn: 6°2′5.11″ E; 49°14′53.65″ N), the Auboué site (Aub: 5°58′33″ E; 49°12′45.7″ N) and the Richemont site (Rich: 6°10′22.2″ E; 49°16′48.1″ N) between November 2019 and February 2021 (the sampling date is given with the site name under the form yy.mm.dd). For the ICP-MS analysis, all samples were filtered to 0.22 µm pore size cellulose acetate and acidified at pH < 2 with HNO_3_ 65% supra pure. The samples were kept at the 6 °C in polystyrene flasks.

For the metal quantification in river water, some samples were measured without any treatment (i.e., unfiltered) and acidified to pH 3.5, while others were filtered and acidified at pH 3.5: (Zn(II), Pb(II) and Cd(II) and some were filtered and acidified at pH < 2: (Pb(II) and Cd(II)). In all cases, a standard addition was carried out by means of adding four aliquots of the pertinent stock solutions. The aliquot volume (between 10 and 50 µL) and stock concentration (10^−4^, 10^−5^ or 10^−6^ M) for each metal ion are chosen as a function of the deposition times used in the SCP measurements.

For comparison, the river samples were also sent to two laboratories: the certified laboratory SARM (Service d’Analyse des Roches et des Minéraux (SARM, CRPG-CNRS UMR 7358, Vandoeuvre-les-Nancy, France) and LIEC). Metal quantification was performed using an inductively coupled plasma atomic mass spectrometer model 7700X Agilent (SARM) and model ICAPTM-TQe-ICP-MS, Thermo Fisher Scientific (LIEC), to compare the estimated concentrations to those obtained using the electrochemical method.

## Figures and Tables

**Figure 1 molecules-26-05502-f001:**
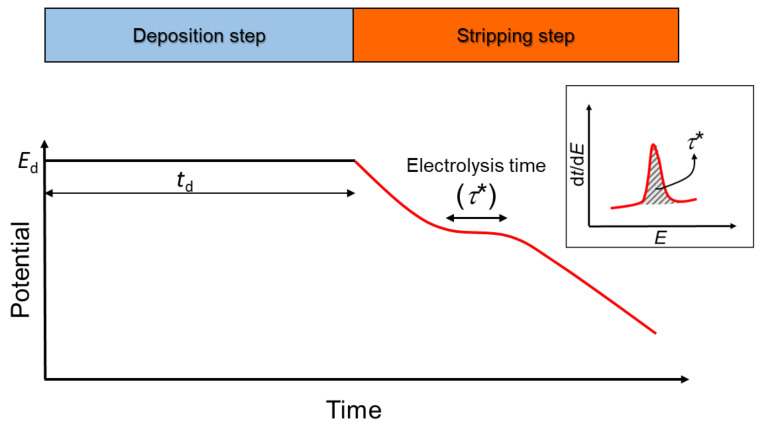
Time-dependence of the potential during stripping chronopotentiometry (SCP). These measurements are conducted using a defined deposition time (*t*_d_ of 45, 90, 180 and 240 s in this work) and a suitable deposition potential (*E*_d_), depending on the metal of interest—normally, −0.75 V for Cd(II) and Pb(II) and −1.35 V for Zn(II) vs. Ag/AgCl [17].

**Figure 2 molecules-26-05502-f002:**
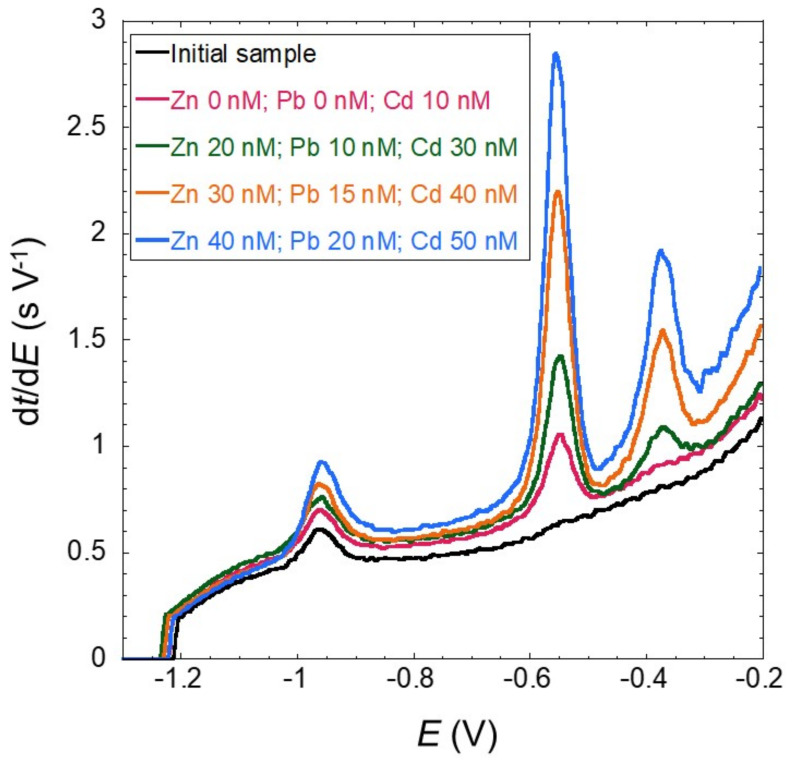
SCP curves obtained for the unfiltered river sample (Aub 19.12.04), (*t*_d_:90 s) and *E*_d_ of −1.25 V at pH 3.5 for the initial sample and standard additions of Cd(II) (−0.55 V peak), Pb(II) (−0.37 V peak) and Zn(II) (−0.95 V peak). Two standard additions were omitted to simplify the data presentation.

**Figure 3 molecules-26-05502-f003:**
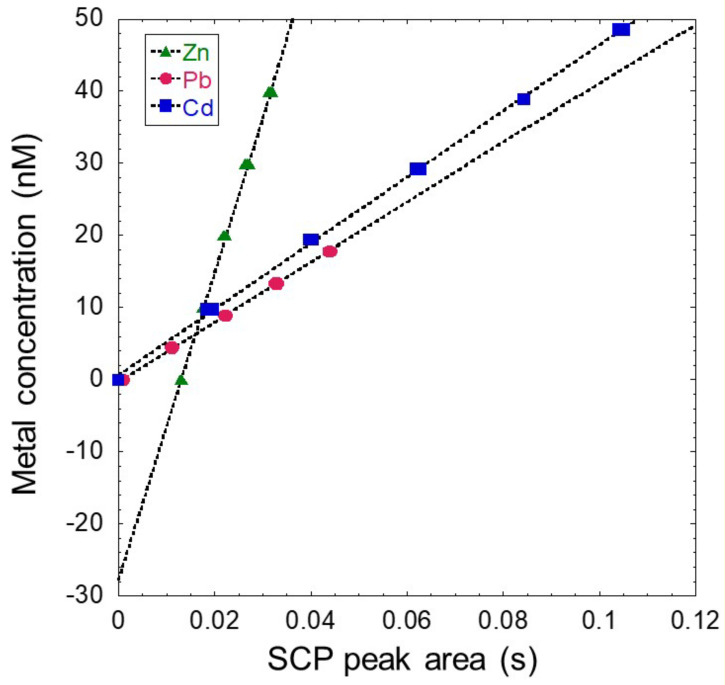
Reversed-axis standard addition method for the unfiltered river sample (Aub 19.12.04), using a *t*_d_ of 90 s and deposition potential (*E*_d_) of −1.25 V at pH 3.5 for Cd(II), Pb(II) (Table 6) and Zn(II) (Table 5).

**Figure 4 molecules-26-05502-f004:**
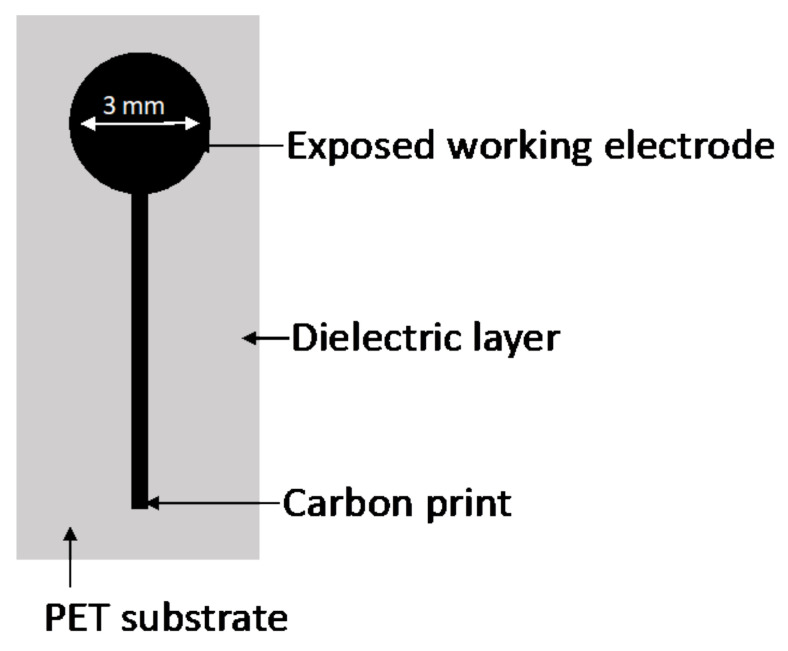
Composition of the screen-printed electrode used. Electrodes comprised a PET substrate, a carbon print and a dielectric layer print. The exposed carbon-paste area was 3 mm in diameter.

**Table 1 molecules-26-05502-t001:** Calibration LODs_y_ for Pb(II), Zn(II) and Cd(II) in 10-mM NaNO_3_ (pH 3.5).

Deposition Time	Zn(II) nM		Cd(II) nM		Pb(II) nM	
(s)	Range	LOD*s*_y_	Range	LOD*s*_y_	Range	LOD*s*_y_
45	7.5–75	1.7	7.5–75	2.2	7.5–75	0.9
45	7.5–75	1.8	7.5–75	2.0	7.5–75	1.0
45	7.5–75	2.4	7.5–75	1.9	7.5–75	1.2
45	7.5–75	2.6	7.5–75	1.8	7.5–75	1.5
45	7.5–75	2.5	7.5–75	1.4	7.5–75	2.3
45	7.5–75	1.8	7.5–75	2.9	7.5–75	2.2
45	7.5–75	1.5	7.5–75	1.1	7.5–75	1.7
45	7.5–75	1.7	7.5–75	1.5	7.5–75	1.8
45	5.0–50	0.9	5.0–50	1.3	5.0–50	1.0
45	5.0–50	1.5	5.0–50	1.5	5.0–50	1.2
45	5.0–50	2.0	5.0–50	1.5	5.0–50	1.0
90	10–50	2.0	10–50	1.1	3.0–15	0.4
90	10–50	1.2	10–50	1.0	3.0–15	0.4
180	-	-	5.0–25	0.4	0.5–4.0	0.08
180	-	-	0.5–2.5	0.03	0.5–4.0	0.07
180	-	-	0.5–3.0	0.04	-	-
180	-	-	0.5–3.0	0.04	-	-
180	-	-	0.5–3.0	0.05	-	-
240	-	-	5.0–25	0.3	0.5–4.0	0.06
240	-	-	0.5–3.0	0.05	-	-
240	-	-	0.5–3.0	0.04	-	-
240	-	-	0.5–3.0	0.06	-	-

**Table 2 molecules-26-05502-t002:** Standard addition LODs for Cd(II) and Pb(II) (*t*_d_: 180 s) in the river water samples at pH 3.5 and at pH < 2.

Sample	pH	Cd(II) nM			Pb(II) nM		
		Addition		LOD*s*_b_	Addition		LOD*s*_b_
Bethdwn 20.02.03	3.5	1.0–2.5	-		0.5–3.0	0.02	
	<2	0.5–2.0	0.03		1.0–4.0	0.03	
Bethdwn 20.02.04	3.5	0.5–2.0	0.04		1.0–4.0	0.04	
	<2	-	-		1.0–4.0	0.04	
Bethdwn 20.02.06	3.5	0.5–2.0	0.02		1.0–4.0	0.04	
	<2	0.5–2.0	0.04		1.0–4.0	0.04	
Bethdwn 20.02.07	3.5	0.5–2.0	0.02		1.0–4.0	0.06	
	<2	-	-		1.0–4.0	0.04	
Aub 20.02.04	3.5	0.5–2.0	0.03		1.0–4.0	0.05	
	<2	0.5–2.0	0.07		1.0–4.0	0.06	
Rich 20.02.04	3.5	0.5–2.0	0.04		1.0–4.0	0.07	
	<2	-	-		1.0–4.0	0.07	

**Table 3 molecules-26-05502-t003:** SCP slopes with standard error and respective slope ratio values for Pb(II), Zn(II) and Cd(II) at 0.1 M NaNO_3_, pH 3.5.

Deposition Time (s)	Pb(II) Slope ×10^5^	Zn(II) Slope ×10^5^	Cd(II) Slope ×10^5^	Pb/Cd Ratio	Zn/Cd Ratio
45	15.43 ± 0.09	5.22 ± 0.03	10.90 ± 0.11	1.41	0.48
45	16.95 ± 0.07	8.45 ± 0.05	12.16 ± 0.08	1.39	0.69
45	11.89 ± 0.05	6.77 ± 0.05	9.58 ± 0.06	1.24	0.71
45	14.15 ± 0.07	8.47 ± 0.07	11.9 ± 0.08	1.19	0.71
45	12.37 ± 0.09	6.77 ± 0.05	9.86 ± 0.05	1.26	0.69
45	10.84 ± 0.08	5.58 ± 0.03	8.79 ± 0.09	1.23	0.63
45	13.01 ± 0.09	7.08 ± 0.03	10.5 ± 0.04	1.24	0.67
45	13.76 ± 0.08	8.28 ± 0.04	10.4 ± 0.06	1.33	0.80
45	11.92 ± 0.08	6.77 ± 0.03	9.23 ± 0.08	1.29	0.73
45	9.56 ± 0.06	5.39 ± 0.04	8.39 ± 0.08	1.14	0.64
45	19.52 ± 0.13	7.60 ± 0.08	12.51 ± 0.10	1.56	0.61
90	24.51 ± 0.18	15.50 ± 0.12	18.49 ± 0.12	1.32	0.84
90	16.10± 0.10	9.82 ± 0.10	14.61 ± 0.09	1.10	0.67

**Table 4 molecules-26-05502-t004:** SCP standard addition slopes and standard errors (deposition time of 90 s for Pb(II), Zn(II) and Cd(II)) in river samples and their ratios at natural ionic strength (acidified to pH 3.5).

Sample	Pb(II) Slope ×10^5^	Zn(II) Slope ×10^5^	Cd(II) Slope ×10^5^	Pb/Cd Ratio	Zn/Cd Ratio
Bethdwn 19.11.29	15.5 ± 0.2	4.14 ± 0.09	13.1 ± 0.2	1.18	0.32
Bethdwn 19.11.30.a	18.3 ± 0.2	4.08 ± 0.08	18.2 ± 0.1	1.01	0.22
Bethdwn 19.11.30.b	20.3 ± 0.2	4.12 ± 0.06	20.2 ± 0.1	1.00	0.20
Bethdwn 19.12.01.a	16.5 ± 0.1	6.36 ± 0.06	16.1 ± 0.2	1.02	0.39
Bethdwn 19.12.01.b	22.5 ± 0.1	9.23 ± 0.09	19.3 ± 0.1	1.16	0.48 *
Bethdwn 19.12.02.a	16.9 ± 0.1	6.25 ± 0.06	18.5 ± 0.2	0.91	0.34
Bethdwn 19.12.02.b	19.3 ± 0.1	6.46 ± 0.10	18.6 ± 0.1	1.04	0.35
Bethdwn 19.12.03	22.9 ± 0.2	5.51 ± 0.08	21.6 ± 0.2	1.06	0.25
Bethdwn 19.12.04	22.0 ± 0.1	4.79 ± 0.06	19.6 ± 0.2	1.12	0.24
Bethdwn 20.02.03	29.6 ± 0.3	6.01 ± 0.10	27.4 ± 0.2	1.08	0.22
Bethdwn 20.02.04	27.8 ± 0.3	7.23 ± 0.11	25.5 ± 0.3	1.09	0.28
Bethdwn 20.02.06	38.5 ± 0.3	7.91 ± 0.07	31.8 ± 0.2	1.21	0.25
Bethdwn 20.02.07	27.7 ± 0.2	7.48 ± 0.08	24.9 ± 0.1	1.11	0.31
Aub 19.11.30	17.8 ± 0.1	5.02 ± 0.03	17.6 ± 0.1	1.01	0.29
Aub 19.12.01	11.7 ± 0.1	3.12 ± 0.06	12.7 ± 0.7	0.92	0.25
Aub 19.12.02	32.7 ± 0.3	7.05 ± 0.11	24.2 ± 0.3	1.35	0.29
Aub 19.12.03	18.9 ± 0.1	1.58 ± 0.05	16.9 ± 0.2	1.12	0.09 *
Aub 19.12.04	24.5 ± 0.1	4.66 ± 0.05	21.8 ± 0.2	1.12	0.21
Aub 20.02.04	26.9 ± 0.2	8.03 ± 0.07	26.3 ± 0.3	1.02	0.31
Rich 20.02.04	41.2 ± 0.4	9.07 ± 0.12	32.7 ± 0.3	1.26	0.28

* Outliers.

**Table 5 molecules-26-05502-t005:** Zn(II) results obtained for and standard addition and internal standard methods (*t*_d_: 90 s) in the river samples (acidified to pH 3.5) and the ICP-MS results obtained in the filtered, strongly acidified samples. a and b written after the sample names refer to replicate measurements.

Sample	Filtered	Zn(II) nM	Zn(II) LOD*s*_b_ nM	Zn(II) Int. Std Zn/Cd; 0.28	ICP-MS (SARM) nM	ICP-MS (LIEC) nM
Bethdwn 19.11.29	No	51 ± 3	4	51 ± 6; 57 ± 11	35 ± 5	-
Bethdwn 19.11.30.a	No	70 ± 2	2	66 ± 5; 54 ± 8	35 ± 5	-
Bethdwn 19.11.30.b	No	76 ± 3	3	68 ± 10; 50 ± 12	35 ± 5	-
Bethdwn 19.12.01.a	No	18 ± 2	2	16 ± 4; 23 ± 7	29 ± 4	-
Bethdwn 19.12.01.b	No	43 ± 2	4	36 ± 4; 62 ± 12	29 ± 4	-
Bethdwn 19.12.02.a	No	34 ± 1	2	25 ± 3; 30 ± 6	51 ± 8	-
Bethdwn 19.12.02.b	No	26 ± 1	2	25 ± 3; 31 ± 6	51 ± 8	-
Bethdwn 19.12.03	No	54 ± 2	2	48 ± 7; 43 ± 9	32 ± 5	-
Bethdwn 19.12.04	No	52 ± 3	2	34 ± 5; 29 ± 7	35 ± 5	-
Aub 19.11.30	No	25 ± 1	2	26 ± 3; 27 ± 5	32 ± 5	-
Aub 19.12.01	No	88 ± 3	3	82 ± 6; 73 ± 10	39 ± 5	-
Aub 19.12.02	No	18 ± 1	3	19 ± 3; 19 ± 5	27 ± 4	-
Aub 19.12.03	No	116 ± 11	7	102 ± 21; 34 ± 8	26 ± 4	-
Aub 19.12.04	No	28 ± 1	2	31 ± 5; 24 ± 5	16 ± 2	-
Bethdwn 20.02.03	Yes	18 ± 2	3	16 ± 2; 13 ± 3	12 ± 2	27 ± 4
Bethdwn 20.02.04	Yes	16 ± 2	3	15 ± 2; 16 ± 3	43 ± 6	38 ± 6
Bethdwn 20.02.06	Yes	17 ± 2	2	16 ± 2; 14 ± 3	32 ± 5	27 ± 4
Bethdwn 20.02.07	Yes	12 ± 1	2	11 ± 1; 12 ± 3	35 ± 5	31 ± 5
Aub 20.02.04	Yes	9 ± 1	2	9 ± 2; 11 ± 3	26 ± 4	33 ± 5
Rich 20.02.04	Yes	17 ± 1	2	18 ± 4; 18 ± 5	31 ± 5	30 ± 5

**Table 6 molecules-26-05502-t006:** Pb(II) results obtained for the standard additions and deposition times of 90 s in the unfiltered river samples at pH 3.5, and their comparison with the ICP-MS results obtained in the strongly acidified samples. a and b written after the sample names refer to replicate measurements.

Sample	Pb(II) nM	Pb(II) LOD*s_b_* nM	Pb(II) Int. Std Pb/Cd; 1.09	ICP-MS (SARM) nM
Bethdwn 19.11.29	2.5 ± 0.5	0.6	-	0.4 ± 0.1
Bethdwn 19.11.30.a	1.8 ± 0.2	0.3	1.7 ± 0.3; 1.6 ± 0.4	0.4 ± 0.1
Bethdwn 19.11.30.b	3.2 ± 0.3	0.4	2.6 ± 0.6; 2.4 ± 0.7	0.4 ± 0.1
Bethdwn 19.12.01.a	0.3 ± 0.1	0.2	0.3 ± 0.1; 0.3 ± 0.1	0.3 ± 0.1
Bethdwn 19.12.01.b	2.1 ± 0.2	0.3	1.8 ± 0.2; 1.9 ± 0.4	0.3 ± 0.1
Bethdwn 19.12.02.a	0.7 ± 0.2	0.3	0.5 ± 0.2; 0.6 ± 0.2	0.3 ± 0.1
Bethdwn 19.12.02.b	0.2 ± 0.2	0.3	0.2 ± 0.1; 0.2 ± 0.1	0.3 ± 0.1
Bethdwn 19.12.03	1.3 ± 0.2	0.3	1.1 ± 0.3; 1.0 ± 0.4	0.6 ± 0.2
Bethdwn 19.12.04	1.1 ± 0.1	0.2	0.7 ± 0.3; 0.7 ± 0.3	0.5 ± 0.2
Aub 19.11.30	1.4 ± 0.2	0.3	1.4 ± 0.2; 1.3 ± 0.3	0.4 ± 0.1
Aub 19.12.01	2.1 ± 0.2	0.3	1.8 ± 0.2; 1.5 ± 0.3	0.6 ± 0.2
Aub 19.12.02	0.5 ± 0.2	0.3	0.8 ± 0.2; 0.7 ± 0.1	0.3 ± 0.1
Aub 19.12.03	1.0 ± 0.2	0.3	0.9 ± 0.2; 0.9 ± 0.2	0.3 ± 0.1
Aub 19.12.04	0.3 ± 0.1	0.2	0.4 ± 0.2; 0.4 ± 0.3	0.2 ± 0.1

**Table 7 molecules-26-05502-t007:** Pb(II) results obtained for the standard additions and internal standard method at a deposition time of 90 s and 180 s in the filtered river samples at pH 3.5 are strongly acidified (<2) and its comparison with the ICP-MS results obtained in the strongly acidified samples.

Sample	Dep. Time/pH	Pb(II) nM	Pb(II) LOD*s*_b_ nM	Pb(II) Int. Std Pb/Cd; 1.09	ICP-MS (SARM) nM	ICP-MS (LIEC) nM
Bethdwn 20.02.03	90 s/3.5	*0.35 < LOD*	0.37	-	0.51 ± 0.17	0.29 ± 0.09
	180 s/3.5	0.11 ± 0.02	0.02	-		
	180 s/<2	0.21 ± 0.02	0.03	0.20 ± 0.05; 0.40 ± 0.13		
Bethdwn 20.02.04	90 s/3.5	1.16 ± 0.22	0.36	1.05 ± 0.15; 1.05 ± 0.10	0.45 ± 0.15	0.53 ± 0.16
	180 s/3.5	0.17 ± 0.02	0.04	0.16 ± 0.01; 0.19 ± 0.02		
	180 s/<2	0.28 ± 0.03	0.04	-		
Bethdwn 20.02.06	90 s/3.5	2.00 ± 0.21	0.29	1.74 ± 0.40; 1.94 ± 0.55	0.33 ± 0.11	0.16 ± 0.05
	180 s/3.5	0.20 ± 0.03	0.04	0.20 ± 0.05; 0.29 ± 0.09		
	180 s/<2	0.10 ± 0.03	0.04	0.09 ± 0.05; 0.18 ± 0.09		
Bethdwn 20.02.07	90 s/3.5	*0.29 < LOD*	0.31	-	0.28 ± 0.09	0.15 ± 0.05
	180 s/3.5	0.42 ± 0.04	0.06	0.43 ± 0.09; 0.69 ± 0.19		
	180 s/<2	0.27 ± 0.03	0.04	-		
Aub 20.02.04	90 s/3.5	0.68 < LOQ	0.24	0.79 ± 0.19; 0.84 ± 0.22	0.32 ± 0.11	0.43 ± 0.13
	180 s/3.5	0.80 ± 0.03	0.05	0.74 ± 0.07; 0.98 ± 0.14		
	180 s/<2	0.35 ± 0.04	0.06	0.28 ± 0.09; *0.80 ± 0.27*		
Rich 20.02.04	90 s/3.5	0.63 < LOQ	0.36	0.83 ± 0.20; 0.96 ± 0.28	0.83 ± 0.28	0.25 ± 0.08
	180 s/3.5	0.31 ± 0.05	0.07	0.33 ± 0.15; 0.52 ± 0.27		
	180 s/<2	0.40 ± 0.05	0.07	-		
Rich 21.02.23	180 s/<2	0.33 ± 0.08	0.10	-	0.21 ± 0.06	-
Aub 21.02.22	180 s/<2	0.29 ± 0.04	0.05	-	0.21 ± 0.06	-
Bethdwn 21.01.26	180 s/<2	0.38 ± 0.06	0.08	-	0.32 ± 0.11	-
Bethdwn 21.01.29	180 s/<2	0.54 ± 0.08	0.10	-	0.16 ± 0.05	-
Bethdwn 21.02.03	180 s/<2	0.41 ± 0.05	0.07	-	0.25 ± 0.08	-
Bethdwn 21.02.05	180 s/<2	0.30 ± 0.04	0.06	-	0.23 ± 0.07	-
Bethdwn 21.02.12	180 s/<2	0.32 ± 0.04	0.05	-	0.14 ± 0.04	-
Bethdwn 21.02.22	180 s/<2	0.34 ± 0.03	0.04	-	0.27 ± 0.08	-

**Table 8 molecules-26-05502-t008:** Cd(II) results obtained for the standard additions (deposition time 180 s) in the filtered river samples at pH 3.5 and strongly acidified (<2) and its comparison with the ICP-MS results obtained in the strongly acidified samples.

Sample	pH	Cd(II) nM	Cd(II) LOD*s*_b_ nM	ICP_MS (SARM) nM
Bethdwn 20.02.03	<2	0.22 ± 0.03	0.03	0.20 ± 0.06
Bethdwn 20.02.06	3.5	0.10 ± 0.02	0.02	0.19 ± 0.06
Bethdwn 20.02.06	<2	0.18 ± 0.03	0.04	0.19 ± 0.06
Bethdwn 20.02.07	3.5	0.14 ± 0.02	0.02	0.06 ± 0.02
Aub 20.02.04	3.5	0.18 ± 0.02	0.03	0.25 ± 0.06
Aub 20.02.04	<2	0.32 ± 0.05	0.07	0.25 ± 0.06
Rich 20.02.04	3.5	0.09 ± 0.03	0.04	0.30 ± 0.09

## Data Availability

The data presented in this study are available on request from the corresponding author.

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
