# Peer review of "Developing On-Site Trace Level Speciation of Lead, Cadmium and Zinc by Stripping Chronopotentiometry (SCP): Fast Screening and Quantification of Total Metal Concentrations"

_molecules, 2021, doi:10.3390/molecules26185502_

Round 1

Reviewer 1 Report

Please check row 136, 138, 176;  missing letters (in my pdf!)

row159-163: 

I suggest to verify the proposed method considering the following paper

A statistical overview of standard (IUPAC and ACS) and new procedures for determining the limits of detection and quantification: Application to voltammetric and stripping techniques
Mocak, J., Bond, A.M., Mitchell, S., Scollary, G., Bond, A.M.
Pure and Applied Chemistry, 1997, 69(2), pp. 297–328

Author Response

Thank you for your time and effort to review our manuscript. We really appreciate it.

The lines mentioned in the reply to the reviewers are those that can be found in the annotated manuscript.

Please check row 136, 138, 176;  missing letters (in my pdf!)

It should be a tau in Greek but it appears in a text font that changes it. I’ve left a commentary for the text editors of the journal in the text of the revised version.

row159-163: 

I suggest to verify the proposed method considering the following paper

A statistical overview of standard (IUPAC and ACS) and new procedures for determining the limits of detection and quantification: Application to voltammetric and stripping techniques
Mocak, J., Bond, A.M., Mitchell, S., Scollary, G., Bond, A.M.
Pure and Applied Chemistry, 1997, 69(2), pp. 297–328

  The method that we have applied corresponds to the RA in the paper above. The application of this method provides very close results as compared with the ULA1 and ULA2, the best methods proposed in this recommendation. This is not surprising since the difference between the RA and ULA1 (that is very similar to ULA2 in most cases) is that the former uses a value of 3 in eq 6 (line 162-163) and ULA1 uses the student t(0.01, n-2). Since n in our case is 15 we obtain a value of 3.012 thus very close to 3. We have added this reference ([19]) and stated that we used an approximation that slightly underestimates the LOD.

Reviewer 2 Report

The subject of the proposed paper is very important about the quality control of water. It was reported that heavy metals belong to water contaminants, with others like pesticides, bacteria, fertilizers, and textile dyes but the list of emerging pollutants increases as a direct consequence of anthropogenic activities in the last decades. There is a lot of studies about the heavy metals content in environmental samples such as soil, plants, or even bee products as an indicator of environmental status. Electrochemical methods such as voltammetry or potentiometry were applied successfully, showing low detection limits of heavy metals. It seems that the water samples are quite simple to analyze, furthermore sometimes it is possible to perform voltammetric analysis without sample pretreatment. The developed instrumentation and methodology used by the authors is not new or innovative being extensively described in the available literature. Moreover, having multiple choice of sensor’s materials (carbon, nanomaterials, conductive polymers), it seems that the proposition of using mercury in the field, despite their excellent electroanalytical properties, is simply unsafe (to put it mildly). In the time of multi-channel sensors, multi-electrode sensor arrays or e-tongues, using thin-mercury film screen printed as a working requires better justification than this given by Authors. The working “portable” FIA system was developed for the analysis of residual contaminants (Zn, Pb) in the water on the International Space Station (ISS) by the way. However, I appreciate this simple approach to solve the in situ problems and showing them feasible. Unfortunately, I do not feel convinced of the proposed methodology of the Cd(II) internal standard.

Comments:

  • The title of the manuscript poorly corresponds with other elements of the work and should be more precise and concise.
  • It seems that the Authors missed some important works from recent years.
  • The objective of the work is not clearly presented. The Authors should be precise about what research is concerning: analysis of selected heavy metal content, speciation of selected metal or analysis of transferable heavy metal ions.
  • Generally, some parts of the manuscript are hard to read with logical, stylistically and typographical errors. Some sections are unnecessary (theory), describing a problem of minor importance, missed essential elements, and some are carelessly written (reagents).
  • The validation of the proposed method is missed or inadequately described. The validation procedure for the proposed method should be extended and described in detail. It seems that the use of modern well-recognized guidelines (i.e. ICH) should be used in this case.
  • The characterization of the working sensor would be appreciated (EIS, SEM)

In my opinion, the manuscript is unsuitable for publication in its current form.

Author Response

Thank you for the revision of our manuscript. We appreciate the effort.

The lines mentioned in the reply to the reviewers are those that can be found in the annotated manuscript.

We have taken this reviewer comment out of order because it is the key aspect in this review and the basis of our reply.

The title of the manuscript poorly corresponds with other elements of the work and should be more precise and concise. The objective of the work is not clearly presented. The Authors should be precise about what research is concerning: analysis of selected heavy metal content, speciation of selected metal or analysis of transferable heavy metal ions.

Effectively in hindsight we could have done a better job of presenting our objectives and organizing this work. Thus, the title was changed to better reflect the manuscript contents:

 “Developing trace level speciation of Lead, Cadmium and Zinc: fast screening and quantification of total metal concentration by Stripping Chronopotentiometry (SCP)”

Then the abstract and introduction were rewritten to clearly describe the objectives and the results and discussion were adapted accordingly.

Briefly our main objective is to perform on-site trace metal speciation at the concentration levels present in natural waters. To achieve this goal, we first need to quantify the total metal concentration, which we describe in this paper.

In the future, the second step will be to quantify the free metal concentration by the technique Absence of Gradients and Nernstian Equilibrium Stripping (AGNES) in the pristine sample using the same electrochemical system (same cell, electrodes and instrument).

The third step will be to investigate the dynamic speciation of the trace metal using the unique properties of Scanned Scanning Chronopotentiometry (SSCP) again using the same electrochemical system. This technique is able to provide insight on the lability and the degree of heterogeneity of the metal complexes present in solution. The SSCP will be carried out in the same sample solution as the free metal determination.

 Since the free metal ion concentration is generally much lower than the total metal the free metal quantification and dynamic speciation studies will not be possible in cases where the total metal concentration is very low, i.e., close or below the LOD for the total metal. Thus, we are interested in developing fast screening techniques that will help to decide for which samples is worthy to carry out the speciation studies and those which the total metal concentration is too low. For this we developed the Cd(II) internal standard methodology.

Our secondary objective was to apply the total metal determination in surface geochemical exploration and use the fast screening methodology to increase the number of samples measured per day, albeit at the cost of increased error on the measured values.

After carefully re-reading the manuscript we got the sense that our secondary objective is more important than the primary objective. This stems from the fact that we’ve measured total metal concentrations being more natural to refer the discussion our secondary objective than to our primary since we actually did not carry out speciation measurements. We have now addressed this problem in the results and discussion.

The developed instrumentation and methodology used by the authors is not new or innovative being extensively described in the available literature.

To the best of our knowledge there is no internal standard method based on Cd(II) to determine Pb(II) and Zn(II). The SCP technique using TMFE on screen printed carbon electrodes is not new, however the LOD presented in this paper for Pb and Cd are the lowest reported for this technique and that opens perspectives of carrying out on-site speciation studies of these metals.

Moreover, having multiple choice of sensor’s materials (carbon, nanomaterials, conductive polymers), it seems that the proposition of using mercury in the field, despite their excellent electroanalytical properties, is simply unsafe (to put it mildly). In the time of multi-channel sensors, multi-electrode sensor arrays or e-tongues, using thin-mercury film screen printed as a working requires better justification than this given by Authors.

If our objective would be only the determination of total metal concentrations then we agree with the reviewer that we could have used other sensor materials. It is the need to perform speciation in the future that imposes mercury films as our working electrodes.  No other electrode material can guarantee the extremely low LOD’s, repeatability and reproducibility of these electrodes.

The electroanalytical properties of mercury are not only excellent they are unique. Being a liquid metal, the surface of any mercury electrode is much smoother than any solid electrode. Additionally the ability to amalgamate a series of environmentally relevant trace metals, Pb(II), Cd(II), Zn(II), Cu(II), Tl(I), In(III) amongst others, coupled with the geometry of the thin films electrodes that have an extremely large area for a very low volume, gives the thin mercury film electrodes the ability to achieve detections limits that are simple not possible with other electrode materials.

The working “portable” FIA system was developed for the analysis of residual contaminants (Zn, Pb) in the water on the International Space Station (ISS) by the way. However, I appreciate this simple approach to solve the in situ problems and showing them feasible.

Unfortunately, I do not feel convinced of the proposed methodology of the Cd(II) internal standard.

The Cd(II) internal standard methodology is intended to be a screening methodology, either to ascertain if it is feasible (and interesting) to carry out speciation studies or to provide fast results when analyzing a lot of samples is required. This point is clarified in the introduction (lines 92-95).

It seems that the Authors missed some important works from recent years.

We believe that this comment is connected with the remark regarding the unclear presentation of the objectives of this work. The reviewer is absolutely correct if our objective would be simply to measure the total metal concentrations in natural systems by electroanalytical techniques. Then of course we did not make a proper review since there are a few hundred papers on the subject with various electrode materials and more or less involved electrochemical cell and different degrees of autonomy.

Nevertheless, our work is constrained to the use of Stripping Chronopotentiometry due to the future goal of performing dynamic trace metal speciation in the samples and to thin film mercury electrodes due to the goal of carrying out the dynamic speciation in-situ. This requires the ability to quantify very low concentrations, sometimes significantly lower than the total metal content with good repeatability, that only mercury electrodes provide.

As we stated in the response to the previous comment we changed the title and rewrote the introduction so that our objectives are clear and in doing so we also revised the references in a way that we hope will satisfy the reviewer.

Generally, some parts of the manuscript are hard to read with logical, stylistically and typographical errors. Some sections are unnecessary (theory), describing a problem of minor importance, missed essential elements, and some are carelessly written (reagents).

We have revised carefully the manuscript regarding the readability, style and typographical errors. We do not agree with the reviewer in that there are unnecessary aspects in the theory section. Maybe the reviewer can specify precisely which are the essential elements that we missed as well as the carelessly written reagents.

The validation of the proposed method is missed or inadequately described. The validation procedure for the proposed method should be extended and described in detail. It seems that the use of modern well-recognized guidelines (i.e. ICH) should be used in this case.

In this paper we are still in the method development stage not in the method validation stage. We have changed the title and the introduction to made this clear and state in the discussion that the method validation will one of the next tasks in our follow up work, but for the whole speciation set-up and not only for the total metal determination.

The characterization of the working sensor would be appreciated (EIS, SEM)

The characterizations of similar sensors have been presented previously by Monterroso et al (ACA, 503, 2004, 203-212) and we have added the proper reference (27) in the materials and methods.

Reviewer 3 Report

In this work, the method based on Stripping Chronopotentiometry (SCP) using electrodeposited thin films of mercury on carbon paste screen-printed electrodes has been developed to quantify subnanomolar concentrations of lead and cadmium and sub-micromolar concentrations of zinc in river water.  The developed method presents great practical significance for geochemistry and environmental chemistry, especially for metal speciations or bioavailabities. Before publication, some improvement are needed for those readers in professional fields reading and then applying. 

A diagrammatic sketch for the improved technique and applying contents could be provided. Analytical processes and special targets? Maybe this will attraction more readers' attention.

The novelty of the manuscript  could be better displayed, such as speciation analysis and transformation process. Maybe in the title or abstract.

The last paragraph should be reorganized and added in its former paragraphs. The results should not be included here.

Table 1 may be transferred into a figure to present good LOD and stablities.

Also some other tables could be transferred to figures for comparison between the proposed method and ICP-MS.

Author Response

Thank you for the revision of our manuscript. We appreciate the effort.

A diagrammatic sketch for the improved technique and applying contents could be provided. Analytical processes and special targets? Maybe this will attraction more readers' attention.

We do not understand what diagram or sketch the reviewer is describing. Can you please clarify.

The novelty of the manuscript could be better displayed, such as speciation analysis and transformation process. Maybe in the title or abstract.

The abstract was rewritten to emphasize the novelty of the paper.

The last paragraph should be reorganized and added in its former paragraphs. The results should not be included here.

After consideration we decided to eliminate this paragraph since it is indeed superfluous.

Table 1 may be transferred into a figure to present good LOD and stabilities. Also, some other tables could be transferred to figures for comparison between the proposed method and ICP-MS.

We are aware of this problem and we have tried to present figures instead of tables but it does not work well at all for our results. All our tables have at least two types of different information, for example Table 1 presents LOD and ranges of calibration and Table 3 and 4 the slopes and the ratios, finally Tables 5-8 the sample measured values and Lod’s. To place this amount and type of information in a figure makes the figure truly unreadable. Also, in some cases the values span more than one order of magnitude like the LOD’s in Table 1 which makes its graphical representation truly problematic.

Reviewer 4 Report

The authors quantified sub-nanomolar concentrations of Pd, Cd and Zn by SCP technique, realized a low LOD range between 10 -100 pM. The results are in good agreement with the LCP-MS technology. Though SCP methodology was reported previously, the authors find approaches to increase the numbers of the analysis pre day. So I think this work is meaningful for the development of SCP methodology. Besides, the experiments were well organized. So I recommend accept this manuscript after minor revision.
1. The sentences are too long to understand, for instance lines 200-204. Please check the whole manuscript and try to utilize a comma to express it clearly.
2. In Figure 1 caption, the Ed should be cited from reference, please cite the references. 
3. In table 5, 6, 7, the significant digits are different, it’s better to use the same significant digits. And for Pb(II), why there are no ICP MS (LIEC) results?

Author Response

Thank you for the revision of our manuscript. We appreciate the effort.

  1. The sentences are too long to understand, for instance lines 200-204. Please check the whole manuscript and try to utilize a comma to express it clearly.
    R: The manuscript was carefully revised. Amongst others changes the mentioned sentence was shortened. 
  1. In Figure 1 caption, the Ed should be cited from reference, please cite the references.

 R: Citation 17 added.

3a. In table 5, 6, 7, the significant digits are different, it’s better to use the same significant digits.

R: The tables were updated and now show the same significant digits.

3b And for Pb(II), why there are no ICP MS (LIEC) results?

R: The results obtained in 2019 present in table 5 and 6 were only measure in the SARM and not in LIEC. Not all samples were sent to both laboratories due to the analysis cost.

Round 2

Reviewer 2 Report

I would like to thank the Authors for considering my suggestions. All my concerns have been addressed. The reagents section should be corrected according to the following suggestions:

  • I have found no reason to use both the chemical formulas and the full names of reagents.
  • p. a. is not the same as Suprapur®
  • The resistivity unit should be changed to 18.2 MΩ cm.
  • The authors claim that standards solutions were from Fluka TraceSELECT® but only reagents are available in the TraceSELECT® line.
  • Which salts were used in metal standard solutions? Were they the standard solutions the same as for ICP/AAS (Fluka, TraceCERT®)? Was the procedure performed as follows: Metallic zinc (high purity quality) in 2% HNO3 (prepared from HNO3 TraceSELECT® and water TraceSELECT®Ultra)?
  • In my opinion, it looks better as follows (example):
    The ionic strength was set using a solution of 1 M of NaNO3 prepared from solid powder (Honeywell Fluka, TraceSELECT®, > 99.999%). The pH was adjusted with 1 M NaOH solution (Merck, Titripur®) or 1 M nitric acid diluted from 65% HNO3 (p.a, Merck, Suprapur®).
    Please consider.

Author Response

Thank you for your time and effort to review our manuscript. Again, we really appreciate it.

We have changed the reagents section according to the referee suggestions.